# Effect of Foot-and-Mouth Disease Virus 2B Viroporin on Expression and Extraction of Mammalian Cell Culture Produced Foot-and-Mouth Disease Virus-like Particles

**DOI:** 10.3390/vaccines10091506

**Published:** 2022-09-09

**Authors:** Victoria Primavera, Janine Simmons, Benjamin A. Clark, John G. Neilan, Michael Puckette

**Affiliations:** 1SAIC, Plum Island Animal Disease Center, Greenport, NY 11944, USA; 2Oak Ridge Institute for Science and Education, Plum Island Animal Disease Center Research Participation Program, Greenport, NY 11944, USA; 3U.S. Department of Homeland Security Science and Technology Directorate, Plum Island Animal Disease Center, Greenport, NY 11944, USA

**Keywords:** FMDV, virus-like particle, vaccine, 2B, viroporin, 3C, antigen, picornavirus, L127P, Foot-and-Mouth disease

## Abstract

To improve the production of foot-and-mouth disease (FMD) molecular vaccines, we sought to understand the effects of the FMD virus (FMDV) 2B viroporin in an experimental, plasmid-based, virus-like particle (VLP) vaccine. Inclusion of the FMDV viroporin 2B into the human Adenovirus 5 vectored FMD vaccine enhanced transgene expression despite independent 2B expression negatively affecting cell viability. Evaluating both wildtype 2B and mutants with disrupted viroporin activity, we confirmed that viroporin activity is detrimental to overall transgene expression when expressed independently. However, the incorporation of 2B into an FMD molecular vaccine construct containing a wildtype FMDV 3C protease, a viral encoded protease responsible for processing structural proteins, resulted in enhancement of transgene expression, validating previous observations. This benefit to transgene expression was negated when using the FMDV 3C^L127P^ mutant, which has reduced processing of host cellular proteins, a reversion resulting from 2B viroporin activity. Inclusion of 2B into VLP production constructs also adversely impacted antigen extraction, a possible side effect of 2B-dependent rearrangement of cellular membranes. These results demonstrate that inclusion of 2B enhanced transgene expression when a wildtype 3C protease is present but was detrimental to transgene expression with the 3C^L127P^ mutant. This has implications for future molecular FMD vaccine constructs, which may utilize mutant FMDV 3C proteases.

## 1. Introduction

Foot-and-mouth disease virus (FMDV), the causative agent of foot-and-mouth disease (FMD), is a member of the Aphthovirus genus in the Picornavirus family. Like other picornaviruses, the FMDV genome encodes a single polypeptide containing structural proteins, comprising the capsid, and non-structural proteins, responsible for viral replication.

Intact capsid structures are required to generate protective immunity from FMD. Molecular FMD vaccines require expression of the P1 polypeptide and the non-structural 3C protease, which processes the P1 into VP0, VP3, and VP1 prior to assembly of virus-like particles (VLPs); peptide VP0 is further processed into VP2 and VP4 by an unknown mechanism upon capsid assembly [1]. A drawback of 3C protease expression is its ability to process multiple host cellular proteins, resulting in detrimental effects on cells and reduced transgene expression [2]. To avoid the deleterious effects of 3C expression, vaccine constructs have incorporated methods to reduce either 3C expression or activity [2,3,4,5,6]. Notable among these is the 3C L127P mutant (3C^L127P^), which reduces the ability of 3C to process multiple host proteins, enhancing transgene expression [2]. 

The FMDV 2B protein is a membrane-bound viroporin with both N- and C-terminal domains extending into the cytosol [7,8,9]. Expression of 2B alters cellular Ca^2+^ concentrations inducing autophagy of host cells [7]. It is unique in containing two distinct pore domains, each able to mediate pore formation, and disrupted by mutations of residues 62 to 65 (2B^ΔA^) or 128 to 130 (2B^ΔB^) [10]. The 2B protein is not known to have a direct role in either processing or assembly of FMDV capsids. However, incorporation into human Adenovirus 5 (Ad5)-vectored FMD vaccines improves transgene expression, potency, and efficacy [11,12]. This enhancement would appear counterintuitive, as independent expression of 2B results in cellular toxicity related to its viroporin function [7,13]. 

Recently a FMDV VLP vaccine platform utilizing mammalian cell culture transiently transfected with plasmids expressing only the P1 and 3C protease was capable of inducing protection from disease in swine [14]. This platform utilized the 3C^L127P^ protease to enhance transgene expression over wildtype 3C (3C^wt^), [2], followed by extraction utilizing a lysis buffer [14]. VLPs produced utilizing this methodology were able to confer protection from challenge with FMDV O1 Manisa in all swine following the administration of two doses [14].

In this report we investigated the effects of 2B on transgene expression independently and upon incorporation into the FMDV VLP producing constructs in mammalian cell culture. Results demonstrate a complicated relationship between 2B expression, viroporin activity, and transgene expression dependent upon 3C activity. We also found the inclusion of 2B altered the ability to extract capsid antigens using established methodologies.

## 2. Materials and Methods

### 2.1. Synthesis of Plasmid Constructs

Genscript synthesized sequences into the pJJP vector which were utilized in this study [15]. Constructs used to evaluate the effect of FMDV O1 Manisa (GI: AY593823) 2B on transgene expression independently utilized the GLuc-Δ1D2A reporter system [16], resulting in plasmid GLuc-2A-2B which was mutagenized by Genscript to produce previously published [10], viroporin activity disrupting mutants, GLuc-2A-2B^ΔA^ and GLuc-2A-2B^ΔB^, as well as double mutant GLuc-2A-2B^ΔA/B^. 

The Δ1D2A-GLucΔ1M reporter system was used to measure the effect of FMDV O1 Manisa 2B on transgene expression when expressed in a VLP producing plasmid [15,16]. The sequence for FMDV O1 Manisa 2B was synthesized and inserted into P1-3C^wt^ and P1-L127P plasmids previously constructed [15], in a way to replicate that of published Ad5 vectors to construct the P1-2B-3C^wt^ and P1-2B-L127P plasmid, respectively. Viroporin domain mutants, P1-2B^ΔA^-L127P, P1-2B^ΔB^-L127P, and P1-2B^ΔA/B^-L127P were produced by mutagenesis of the P1-2B-L127P plasmid by Genscript.

For independent expression, the O1 Manisa 2B sequence was synthesized into the pJJP plasmid without a luciferase reporter. The sequence of O1 Manisa VP1 was also synthesized into the pJJP plasmid for use as a control. The construct A24P1-L127P was constructed by synthesis of the P1-2A from FMDV A24 Cruzeiro (GI: AY593768) and replacement of the O1 Manisa P1-2A sequence in pJJP P1-L127P.

### 2.2. Transfection of Cell Cultures

#### 2.2.1. Transfection of Cell Cultures with Single Plasmids

HEK293-T (ATCC, CRL-11268) and CHO-K1 (ATCC, CCL-61) cell cultures were transfected with single plasmids utilizing Lipofectamine 2000, as per manufacturers suggestions (Thermo Fisher Scientific, Waltham, MA, USA). Transfected cells were maintained at 37 °C and 5% CO_2_ for 24 h before media was harvested for luciferase assays.

#### 2.2.2. Transfection of Cell Cultures with Two Plasmids

For transfection of two plasmids simultaneously, HEK293-T cells were cultured in six-well plates and simultaneously transfected with 2 μg of each plasmid utilizing Lipofectamine 2000 (Thermo Fisher Scientific, Waltham, MA, USA) as per manufacturers suggestions. Transfected cells were maintained at 37 °C and 5% CO_2_ for 24 h after which media was harvested for use in luciferase assays and cells lysed with M-PER (Thermo Fisher Scientific, Waltham, MA, USA) for antigen detection.

### 2.3. Luciferase Assay

Luciferase assays were performed as previously described [15]. In brief, cell culture supernatant from transiently transfected cell cultures were harvested 24 h post-transfection and clarified through centrifugation at 500× *g* for 3 min. Clarified media was diluted to a 1:4 dilution to prevent signal over-saturation and evaluated for luciferase activity utilizing a 96-well BioSystems Veritas luminometer (Turner Biosystems, Sunnyvale, CA, USA) with 100 μL of diluted sample in each well. Readings were taken following injection of 100 μL of 50 μg water-soluble coelentrazine (Nanolight Technology, Pinetop, AZ, USA) using an integration time of 0.5 s both before and after injection of substrate.

### 2.4. Electron Microscopy

The formation of arrays comprised of FMDV VLPs produced using this system had been previously established using immuno-electron microscopy [16]. For TEM processing, cells were fixed in 2.5% glutaraldehyde, 2% paraformaldehyde in 0.1 M sodium cacodylate buffer, postfixed with 1% osmium tetroxide, stained en bloc with 2% uranyl acetate, dehydrated through a graded series of ethanol then propylene oxide, and embedded in Spurr’s resin (Electron Microscopy Sciences, Hatfield, PA, USA). Ultrathin (80 nm) sections were cut on a Leica UC6 microtome, post-stained with uranyl acetate and lead citrate, and imaged on a Hitachi 7600 TEM with a 2k × 2k AMT camera at 80 kV.

### 2.5. Harvest and Detection of Antigen from Transfected Cells

#### 2.5.1. Transfection of Cell Cultures for Antigen Extraction

Transient transfection of cell cultures with DNA plasmid for antigen production was performed using transfection grade polyethylenimine MW 25,000 (Polysciences, Warrington, PA, USA) as previously described [14]. Following transfection, cells were incubated at 37 °C and 5% CO_2_ overnight prior to harvest. Antigen extraction was performed utilizing lysis buffers, LB9 (20 mM Tris-HCl, 200 mM NaCl, 3 mM MgCl_2_, 0.1% TritonX-100), LB9A (20 mM Tris-HCl, 200 mM NaCl, 3 mM MgCl_2_, 0.5% TritonX-100), LB9D (20 mM Tris-HCl, 200 mM NaCl, 3 mM MgCl_2_, 0.1% TritonX-100, 0.1% SDS), or RIPA (25 mM Tris-HCl, 150 mM NaCl, 1% NP-40, 1% sodium deoxycholate, 0.1% SDS) (Thermo Fisher Scientific, Waltham, MA, USA).

In brief, cells were removed by flushing with cell culture media and transferred to 15 mL conical tubes. Cells were pelleted by centrifugation at 500× *g* for 5 min, and the supernatant discarded. Cells were washed by resuspension in 1 mL of dPBS and pelleted by centrifugation at 500× *g* for 5 min. Cell pellets were resuspended in lysis buffer, either LB9, LB9A, LB9D, or RIPA, and incubated for 10 min at room temperature on a shaker to lyse cells. Following incubation, cellular debris was removed by centrifugation at 500× *g* for 5 min, and supernatant was retained for western blotting. For cells lysed with RIPA, 1 μL of DNase I (Sigma-Aldrich, St. Louis, MO, USA) was added, and the mixture was incubated at 37 °C for 1 h.

#### 2.5.2. Detection of Extracted Antigen by Western Blotting

Western blotting was performed as previously described [2], utilizing transfected HEK293-T (ATCC, CRL-11268) cells with loading equilibrated using RFUs/0.5 s from GLuc expression when applicable and utilizing the following primary antibodies, rabbit polyclonal antisera GLuc (#401P, Nanolight Technology, Pinetop, AZ, USA) at 1:1000 dilution, F1412SA [17], at 1:10 dilution, VP3-444-9 at 1:10 dilution, 12FE9.2.1 [18], at 1:10 dilution, Clone 38 [19], at a 1:1 dilution, and 1F1E4 at 1:10 dilution. Secondary antibodies used at a 1:500 dilution were goat anti-rabbit-HRP for GLuc polyclonal, goat anti-mouse-HRP for F1412SA, VP3-444-9, 12FE9.2.1 and 1F1E4, and goat anti-swine-HRP for Clone 38. Chromogenic detection was performed using either SIGMAFAST™ 3,3′-Diaminobenzidine (Sigma-Aldrich, St. Louis, MO, USA) or 1-Step™ TMB-Blotting Substrate Solution (Thermo Fisher Scientific, Waltham, MA, USA).

## 3. Results

### 3.1. Expression of 2B Had a Negative Effect on Transgene Expression

We utilized the GLuc-Δ1D2A reporter system, Figure 1A, to evaluate transgene expression by 2B viroporin activity using either single trans-membrane domain alanine mutants, 2B^ΔA^ or 2B^ΔB^, or a mutation of both trans-membrane domains, 2B^ΔA/B^. In both HEK293-T and CHO-K1 transfected cells, all three mutants enhanced transgene expression over 2B, and mutant 2B^ΔB^ produced the greatest enhancement, Figure 1B.

### 3.2. Enhancement of Transgene Expression by 2B Was Dependent upon 3C Activity

Plasmid vectors expressing P1-2A, 3C, and the Δ1D2A-GLucΔ1M reporter, Figure 2A, were previously constructed to evaluate VLP assembly and transgene expression [15]. For this study we incorporated a 2B containing sequence mirroring that of previous Ad5 vectors, Figure 2B. The incorporated sequence contained 2B and additional sequences derived from FMDV non-structural proteins 2C, 3A, and 3B, Figure 2C. 

Incorporation of 2B into VLP constructs expressing 3C^wt^ enhanced transgene expression in both HEK293-T and CHO-K1 cell lines, Figure 3A. In contrast, incorporation of 2B into VLP constructs containing the 3C^L127P^ mutant decreased transgene expression, Figure 3B. This decrease was partially abated by mutation of 2B viroporin domains, Figure 3B. To evaluate if 2B reduction of transgene expression with 3C^L127P^ was the result of an altered construct design we expressed 2B independently of P1-L127P using a two-plasmid system. In this system, only the P1-L127P plasmid contained a luciferase reporter. While expression of P1-L127P was within the standard deviation of that observed when VP1 was expressed from a second plasmid, a substantial decrease in expression was observed when 2B was expressed, Figure 3C, suggesting that the reduced transgene expression caused by 2B expression with 3C^L127P^ was independent of construct design. This 2B-dependent decrease was replicated with a plasmid encoding the P1 polypeptide from FMDV serotype A strain Cruzeiro, A24P1-L127P, demonstrating independence from the FMDV O1 Manisa P1 polypeptide, Figure 3D.

### 3.3. Incorporation of 2B in Plasmids Did Not Affect the Presence of VLP Arrays in Transfected Cell Culture

Previous utilization of the P1-L127P plasmid demonstrated the formation of VLP arrays within transfected cells [2,15,16]. Studies using the Ad5 vaccine did not observe arrays in infected cells but did demonstrate an increase in cytoplasmic vesicles following 2B incorporation [11]. In this study, VLP arrays were observed independent of 2B inclusion, Figure 4. In cells transfected with the P1-2B-L127P construct many VLP arrays, but not all, were fully encircled by membrane vesicles, Figure 4. No arrays or similar structures were observed in cells transfected with constructs containing a 3C^C163A^ protease mutant [2,15], which is unable to process the P1 (data not shown).

### 3.4. Inclusion of 2B Inhibits Antigen Extraction

Antigen from transfected cell cultures were extracted utilizing LB9 and evaluated by western blotting. Antigens were detected with multiple monoclonal antibodies using the P1-L127P extract but not detected with P1-2B-L127P extract, Figure 5A. This effect was replicated utilizing a two-plasmid transfection system, Figure 5B. 

Lysis of cells with buffers containing either more TritonX-100, LB9A, or SDS, LB9D and RIPA, did not fully counteract the observed effect, although use of RIPA buffer for extraction revealed possible low molecular weight degradation products, Figure 6. These products were detected with two different VP0/2 antibodies, F1412SA and 1F1E4. Interestingly no impairment of GLuc detection by western blot was observed, Figure 5A and Figure 6.

## 4. Discussion

Expression of either FMDV 2B viroporin or 3C protease in mammalian cells is detrimental to cellular activity and transgene expression [2,7]. In our studies, the negative effect of 2B on transgene expression was related to its viroporin activity, and mutations that removed or reduced this activity resulted in enhancement of transgene expression, Figure 1B. Published results documenting that inclusion of 2B enhances FMD vaccine expression, potency, and efficacy were performed using the Ad5 vector system expressing a 3C^wt^ protease [11,12]. We also found that when 2B was incorporated into constructs utilizing 3C^wt^, transgene expression was enhanced, Figure 3A, in agreement with Ad5 reports [12]. Our results agree with both reports of 2B expression having harmful effects on cells, and its inclusion enhancing FMD vaccine expression. This increase could result in improved Ad5 vaccine potency and efficacy, independent of other potential alterations caused by the presence of 2B. 

Expression of FMDV 3C^wt^ is deleterious to both host cells and transgene expression due to its ability to process host cellular proteins, however this impact can be negated by utilizing the 3C^L127P^ mutant [2]. Multiple FMD molecular vaccine platforms incorporate strategies to mitigate 3C activity for enhancement of expression, such as IRES [3,4], HIV frameshift [6], or 3C mutagenesis [4,5,6,15]. However, it is unknown how incorporation of 2B would influence transgene expression for each strategy. Therefore, the relationship between 2B, 3C, and transgene expression reported here suggest multiple possibilities and demonstrate the critical need to evaluate the effects of including 2B in each circumstance.

A possible reason for the beneficial effects of 2B with 3C^wt^ is the design of the constructs. The 2B-containing sequence utilized here and previously [11,12], contains non-2B-related sequences, Figure 2C, including junctions processed by 3C. Incorporation of these potential 3C processing sites may play a role in the enhancement seen with 3C^wt^ constructs by providing additional substrate for the protease. In the Ad5 vectored vaccines, the addition of 2B with 94 truncated residues along with 3B removed these potential 3C processing sites and did not enhance transgene expression [12]. While a reduction in expression was observed with P1-2B-L127P relative to P1-L127P, expression was still greater than that observed with P1-2B-3C^wt^. Inclusion of both 3C^L127P^ and 2B in an Ad5 vectored FMD vaccine resulted in full protection in cattle [2], demonstrating that 3C^L127P^ and 2B remain compatible with inducing protection from clinical disease.

Inclusion of 2B is not necessary for the construction of virus like particles [2,6,15,16], and the work presented here demonstrated the presence of VLP arrays in transfected cells without 2B, Figure 4. VLPs produced using P1-L127P constructs elicit protection from clinical FMD in both cattle and swine [14]. The same extraction methodology applied to P1-2B-L127P-expressing cells resulted in undetectable antigens in western blots with multiple antibodies, Figure 5A. This effect was duplicated using a two-plasmid expression system with P1-L127P, Figure 5B, demonstrating its independence of the P1-2B-L127P construct and dependent upon 2B expression. 

Significant membrane rearrangements are reported following 2B expression [7,11]. While not a universal effect, multiple VLP arrays observed in P1-2B-L127P-transfected cells were encapsulated within membrane vesicles, Figure 4. It is unknown if this was due to an interaction of VLPs, or precursors, with 2B or whether 2B-induced membrane rearrangement created opportunities for the occurrence. 

Interestingly, antigen could not be readily extracted from P1-2B-L127P-expressing cells using the LB9 lysis methodology, Figure 5. Low levels of LB9 extracted antigen were visible by using 1-Step™ TMB-Blotting Substrate Solution, Figure 6. To enhance lysis of transfected cells, lysis buffers containing alternative detergent formulations were utilized. While these buffers may enhance extraction to a minor degree, they were unable to produce major changes in visualization of the extracted antigen, Figure 6. These results demonstrated that the previous LB9-based methodology of VLP extraction was ineffective when 2B was expressed. This effect appears independent of viroporin activity, although it is possible that limited residual activity is retained in viroporin mutants. 

The use of extraction buffers LB9D and RIPA, containing the detergent SDS, resulted in the presentation of some lower molecular weight bands around 14 kDa, possibly representative of degradation products, Figure 6. As 2B is known to induce autophagy in expressing cells, these bands may represent degradation products of VP0 or VP2 resulting from autophagy within the transfected cell.

## 5. Conclusions

Incorporation of 2B into FMDV VLP constructs had different effects on transgene expression dependent upon the 3C protease sequence incorporated. Utilizing a 3C^wt^ construct resulted in inclusion of 2B enhancing transgene expression, while 3C^L127P^ constructs had reduced transgene expression in the presence of 2B. The addition of 2B into 3C^L127P^ constructs severely limited the ability to extract capsid antigens detectable by western blotting. It remains unknown if the VLPs produced by 2B-containing constructs are structurally or immunologically different and whether this can influence vaccine potency or efficacy. These findings demonstrate the need to define circumstances in which inclusion of 2B will be beneficial to FMD VLP vaccine production, potency, and efficacy. 

## Figures and Tables

**Figure 1 vaccines-10-01506-f001:**
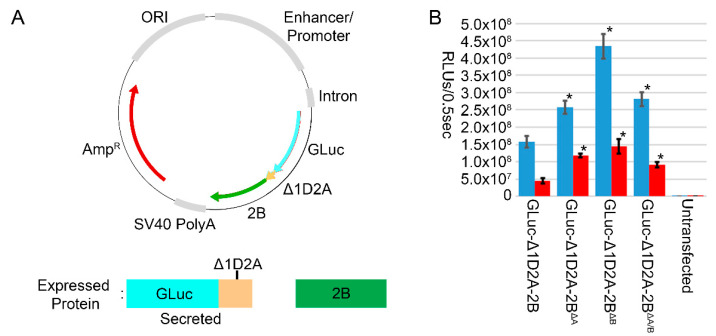
(**A**) The GLuc-Δ1D2A reporter system used to evaluate 2B effects on transgene expression. The plasmid encoded GLuc-Δ1D2A and 2B are expressed as a single polypeptide, upon translation ribosome skipping was induced resulting in separate expression of GLuc-Δ1D2A and 2B. The GLuc-Δ1D2A was secreted into the media where it was quantified in relative luciferase units per half second (RLUs/0.5 s). (**B**) Luciferase activity for GLuc-2A-2B constructs with and without viroporin-disrupting mutations, in transfected cell cultures: HEK293-T (blue) and CHO-K1 (red). * denotes a significant difference, *p* < 0.001 as determined by Student’s T-test, between expression of wildtype 2B and viroporin mutants.

**Figure 2 vaccines-10-01506-f002:**
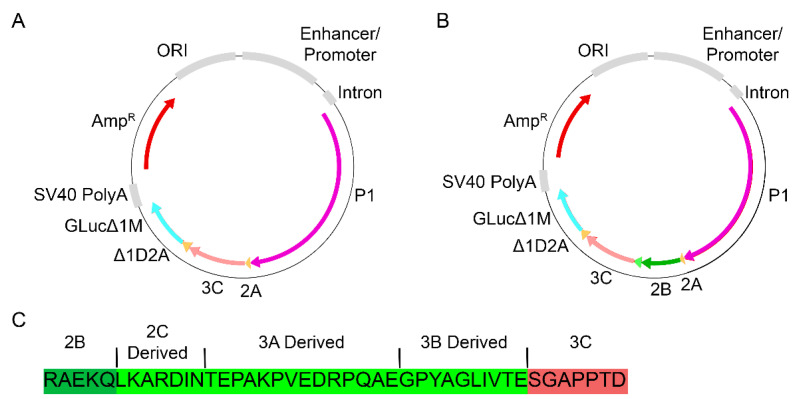
VLP-producing constructs expressed as a single polypeptide (**A**) P1, 2A, 3C, and the Δ1D2A-GLucΔ1M reporter or (**B**) P1, 2A, 2B, 3C, and the Δ1D2A-GLucΔ1M reporter. (**C**) Incorporation of 2B into the VLP construct mimicked previous Ad5 FMD vaccines containing 2B in addition to amino acids derived from 2C, 3A, and 3B.

**Figure 3 vaccines-10-01506-f003:**
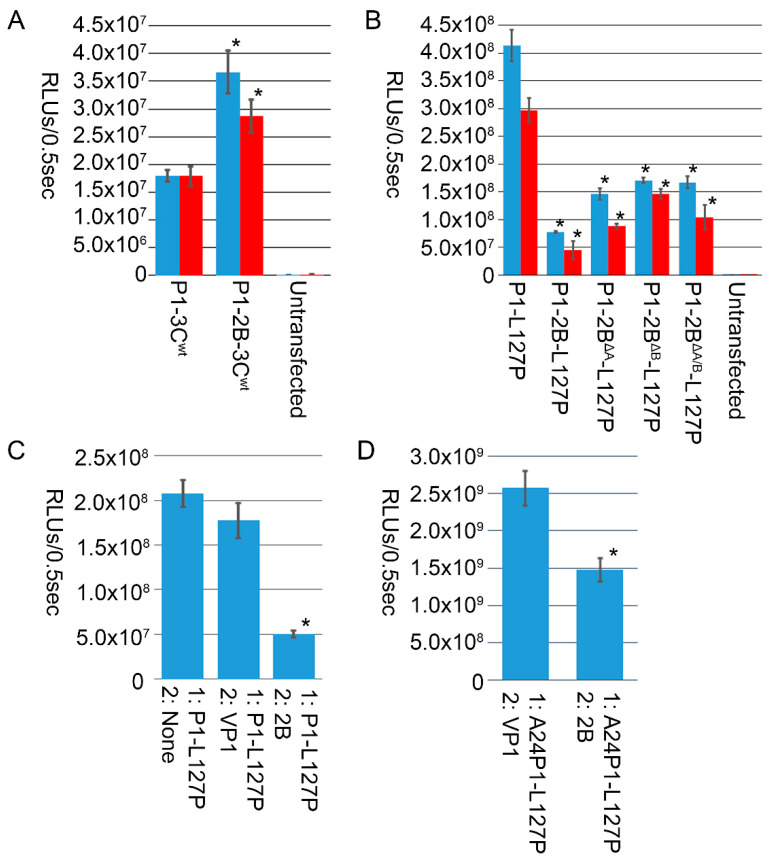
Luciferase activity, in RLUs/0.5 s, for HEK293-T (blue) or CHO-K1 (red) cells transfected with, (**A**) constructs containing P1 and either non-structural proteins 3C^wt^ alone, P1-3C^wt^, or both 2B and 3C^wt^ combined, P1-2B-3C^wt^, demonstrated enhancement of transgene expression with inclusion of 2B. (**B**) Constructs containing P1 and either non-structural proteins 3C^L127P^ alone, P1-L127P, or both 2B and 3C^L127P^, P1-2B-L127P, demonstrated a reduction in transgene expression with inclusion of 2B. This decrease in transgene expression was partially negated in constructs containing mutated viroporin domains. (**C**) Simultaneous transfection of two plasmids resulted in a decrease in luciferase activity with a 3C^L127P^ construct in the presence of 2B, demonstrating that reduction of transgene expression was independent of construct design. (**D**) This decrease was replicated with a P1 corresponding to the A24 serotype. * denotes a significant difference, *p* < 0.001 as determined by Student’s T-test, between expression of constructs with and without wildtype 2B or 2B viroporin mutants.

**Figure 4 vaccines-10-01506-f004:**
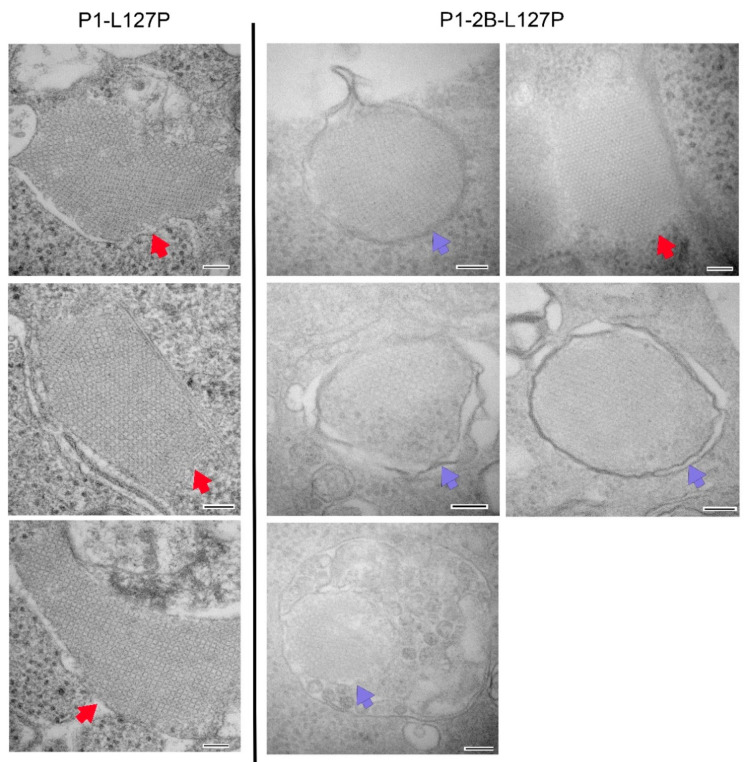
TEM images of VLP arrays within HEK293-T cells transfected with plasmids P1-L127P or P1-2B-L127P. In cells transfected with P1-2B-L127P, multiple arrays were found encapsulated by membrane vesicles. Scale bar represents 100 nm. Red arrows point to VLP arrays not definitively encapsulated by membrane structures while blue arrows point to VLP arrays observed encapsulated by membranes.

**Figure 5 vaccines-10-01506-f005:**
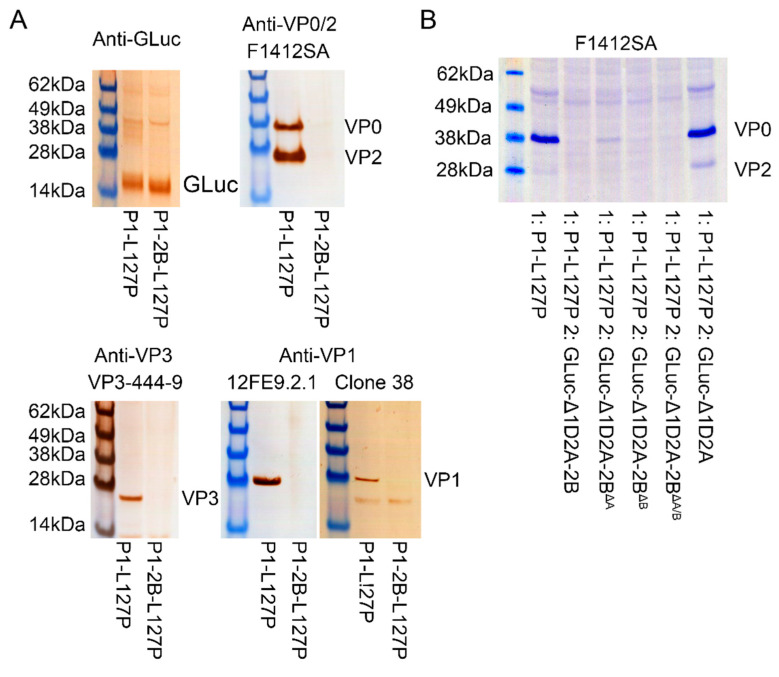
Western blotting of antigens extracted with LB9 from (**A**) cells transfected with P1-L127P or P1-2B-L127P demonstrate a lack of detectable antigen with P1-2B-L127P. (**B**) To ensure that this effect was dependent on 2B expression and not a result of the P1-2B-L127P plasmid simultaneous transfection of P1-L127P with and without 2B, or 2B mutant, expressing plasmids replicated the effect.

**Figure 6 vaccines-10-01506-f006:**
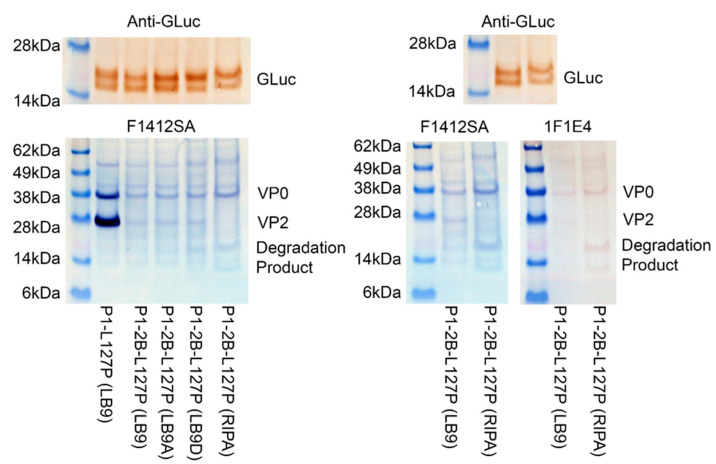
Utilization of alternative lysis buffers did not counter the adverse effect of 2B expression on antigen extraction. Detection with two different VP0/VP2 antibodies, F1412SA and 1F1E4, also demonstrated the presence of low-molecular-weight products, possibly indicative of VP0/VP2 degradation in 2B-expressing cells.

## Data Availability

All datasets produced and analyzed for this study are available from the corresponding author on reasonable request.

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
