# Peer review of "Effect of Foot-and-Mouth Disease Virus 2B Viroporin on Expression and Extraction of Mammalian Cell Culture Produced Foot-and-Mouth Disease Virus-like Particles"

_vaccines, 2022, doi:10.3390/vaccines10091506_

Round 1

Reviewer 1 Report (New Reviewer)

Foot-and-mouth disease virus (FMDV) is the causative agent of Foot-and-Mouth disease, to explore the effect of virus protein 2B in production of VLP vaccine, the authors include the 2B into the human Adenovirus 5 vectored FMD vaccine and found that incorporation of 2B into FMD molecular vaccine construct resulted in enhancement of transgene expression, and more interestingly, the authors demonstrated a complicated relationship between 2B expression, viroporin activity, and transgene expression dependent upon FMDV 3C activity. The experiments were well designed, and data clearly showed, overall this study is of a potential interests to the experts in the field. The minor concerns of this work are shown as below:

1. General remark: Almost all figures should be well organized, like Fig 3, what is the red and blue bar representing for? the X-axis are so crowded and need cut off some numbers to make it concise; like Fig 5 and Fig 6, the protein molecular weight kDa should be labeled on the left top of each picture instead of label each lane. TEM pictures in the Figure 4, the qualities is not good and not some is smear.   In addition, Please give the explanation of pics of Figure 4 and point out VLPs with arrow,  otherwise it is hard to understand what they are.

2. Can authors specify in the abstract that 3C protein is the viral protein 3CPro or else?

3. Does the inclusion of 2B or 3C affect cell viability?

4. English editing is required.

Author Response

  1. General remark: Almost all figures should be well organized, like Fig 3, what is the red and blue bar representing for? the X-axis are so crowded and need cut off some numbers to make it concise; like Fig 5 and Fig 6, the protein molecular weight kDa should be labeled on the left top of each picture instead of label each lane. TEM pictures in the Figure 4, the qualities is not good and not some is smear.   In addition, Please give the explanation of pics of Figure 4 and point out VLPs with arrow,  otherwise it is hard to understand what they are.
  • The purpose of the red and blue bars in Figure 3 is to represent performance of the assay in different cell lines, HEK293-T (blue) and CHO-K1 (red).  Definition of this is included in the figure caption.
  • Added arrows pointing to VLP arrays in Figure 4, red arrows designate arrays not observed as encapsulated by membranes while blue arrows point to those observed encapsulated by membranes.  Added a section in the figure description for Figure 4 defining difference between red and blue arrows.
  1. Can authors specify in the abstract that 3C protein is the viral protein 3CPro or else?
  • Added in the abstract, line 19-20, that the wildtype FMDV 3C protease is a viral encoded protease  responsible for processing structural proteins.
  1. Does the inclusion of 2B or 3C affect cell viability?
  • Yes, expression of either active 2B or active 3C has a negative effect on cell viability.
  1. English editing is required.
  • Changes have been made to the body of the manuscript

Reviewer 2 Report (New Reviewer)

  How to improve FMD VLP vaccine production, potency, and efficacy is very important and necessary. In this paper, authors we investigated the effects of 2B on transgene expression independently and upon incorporation into FMDV VLP producing constructs. Results demonstrated a complicated relationship between 2B expression, viroporin activity, and transgene expression dependent upon 3C activity. it also found the inclusion of 2B altered the ability to extract capsid antigens using established methodologies.

    In general, results are convincingly presented and constitute an interesting contribution to the FMDV VLP vaccine.

Specific issues:

1.      Results 3.1, Fig.1B, How to ensure the consistency between the experimental results of luciferase assay and transgene expression?

2.      Fig.3, How to ensure the consistency between the experimental results of luciferase assay and VLP in HEK293-t AND CHO-K1 cells? It should be shown that 3C mutant affected it’s activity.

3.      Differences should be marked with ※ in Fig.1B and Fig.3

4.      “FMD vaccine” in the last sentence in “part 4, conclusions” should be replaced with “FMD VLP vaccine” or “FMD genetical engineered vaccine”.

5.      The present title of this manuscript is too broad, conclusion should be reflected in the title, please modify.

Author Response

  1. Results 3.1, Fig.1B, How to ensure the consistency between the experimental results of luciferase assay and transgene expression?
  • The concern of consistency between experimental results utilizing the luciferase assay is of great importance to the authors. The authors utilize the following to enhance consistency and account for variations that may influence conclusions drawn from results:
    • All plasmids utilized were purchased from a manufacturer (Genscript) to enhance the quality and purity of the plasmids. This reduces the risk of transfecting cells with samples containing non-plasmid DNA.
    • Comparisons between samples is performed on samples transfected, harvested, and assayed at the same time. This allows for better controlling of variables such as differences in cell densities at the time of transfection. All data presented in each discrete graph presenting luciferase data was obtained at the same time.
    • Results are obtained from three unique transfections of cell cultures performed in parallel. Media harvested from each transfection is subsequently assayed utilizing between 6-7 replicates prior to pooling the data together.
    • Substrate, water soluble coelenterazine (NanoLight Technology) was resuspended fresh for each assay. Each tube provided by the manufacturer is sufficient for a single 96 well plate.  If multiple 96 well plates were needed for an experiment multiple tubes were resuspended and pooled together prior to running the assay to account for any possible differences that might emerge due to the substrate.
  1. Fig.3, How to ensure the consistency between the experimental results of luciferase assay and VLP in HEK293-t AND CHO-K1 cells? It should be shown that 3C mutant affected it’s activity.
  •  The authors have published and reference previous work performed looking at the effect of 3C on transgene expression in HEK293-T and CHO-K1 cell lines, Puckette et al 2017 and Martel et al 2019.  Martel et al 2019 in particular looks at multiple 3C mutants in both cell lines and how that compares to expression of both wildtype and inactivated 3C.

  1. Differences should be marked with ※ in Fig.1B and Fig.3
  • Added * to denote p < 0.001 as determined by Student T-test in Figure 1B and Figure 3. Made not of * in figure description.

  1. “FMD vaccine” in the last sentence in “part 4, conclusions” should be replaced with “FMD VLP vaccine” or “FMD genetical engineered vaccine”.
  • Added VLP between FMD and vaccine 
  1. The present title of this manuscript is too broad, conclusion should be reflected in the title, please modify.
  • Changed title to focus on expression and extraction of VLPs

Reviewer 3 Report (New Reviewer)

- The authors presented interesting findings about FMDV 2B viroporin as it related to VLP vaccines. The experiments were well thought out and the data supports the conclusions present in the manuscript. The only comments and suggestions I have are the following: 

- The manuscript would benefit from the addition of more detail though out. Specifically, the introduction/background and the material and methods sections would benefit from more detail. 

- It would be useful to include a brief description of the methods used in the luciferase assay (lines 94/95) rather than stating assays were performed as previously described 

- The overall flow of the background/introduction section could be improved for ease of understanding, especially if the audience is not familiar with FMDV VLPs. 

- Is there any understanding how inclusion of 2B impacts the efficacy of an FMDV VLP vaccine

- What will the ultimate benefit be if inclusion of 2B in the mutant construct reduced transgene expression?   

Author Response

- The manuscript would benefit from the addition of more detail though out. Specifically, the introduction/background and the material and methods sections would benefit from more detail. 

  • The authors have expanded sections of the materials and methods, specifically the luciferase assay as requested by this and other reviewers.  The authors have also expanded the introduction to include more information of the VLP platform utilized in this study.

- It would be useful to include a brief description of the methods used in the luciferase assay (lines 94/95) rather than stating assays were performed as previously described 

  • Added more in depth description

- The overall flow of the background/introduction section could be improved for ease of understanding, especially if the audience is not familiar with FMDV VLPs. 

  • The authors have also expanded the introduction to include more information of the VLP platform utilized in this study.

- Is there any understanding how inclusion of 2B impacts the efficacy of an FMDV VLP vaccine

  • There is not.  Previous reports of enhanced efficacy following inclusion of the vaccine do not determine if this enhancement is the result of 2B activity on generated antigen or the result of enhanced transgene expression resulting in more antigen being present.  It should be noted that previous work including 2B was performed using the Ad5 vaccine system which is not a VLP vaccine but rather produces empty capsid within a hosts cells following vaccination.

- What will the ultimate benefit be if inclusion of 2B in the mutant construct reduced transgene expression?   

  • There does not appear to be a benefit when utilized with the L127P protease.  The authors undertook this research to determine if inclusion of 2B would enhance the mammalian VLP vaccine platform recently developed, Puckette et al 2022, as has been previously reported for the Ad5 platform, Pena et al 2008 and Moraes et al 2011.  The authors have concluded from this work that inclusion of 2B into the expression construct is not of benefit to the mammalian VLP platform.

Reviewer 4 Report (New Reviewer)

Primavera and colleagues have investigated the effects of 2B on GLUC reporter expression and upon incorporation into FMDV VLP producing constructs containing either wildtype 3C or 3CL127P mutant. Although the work is interesting, I feel further experiments are required to support the conclusions. 

The authors should clarify the method used for luciferase assays. The authors state in the materials and methods section that “Luciferase assays were performed as previously described [15].” This reference states “Luciferase activity was detected in media separated from transfected cells”.  Was this the method used?

With regards to the data presented in Figure 1B, is it possible that expression of wild type 2B affects GLuc (transgene) secretion into the medium rather than expression per se? Why does the western blot of cell lysates in Figure 5A show comparable GLuc expression for both P1-L127P or P1-2B-L127P? – these results appear inconsistent with those presented in Figure 3B which show “a reduction in transgene expression with inclusion of 2B”. Likewise, in Figure 6 the amount of GLuc expressed in the absence or presence of 2B appear comparable when cell lysates are analysed by western blot.

The authors show that the expression of 2B prevents antigen extraction using the listed lysis buffers. Have the authors analysed non-clarified whole cell lysates to determine if the antigen is pelleted with the cellular debris when 2B is expressed?

Line 153: The authors should clarify the function of the additional sequences derived from FMDV non-structural proteins 2C, 3A, and 3B.

Line 160, and Figures 3A and 3B: It’s difficult to compare the results presented in Figures 3A and 3B if they were not carried out at the same time. The authors should repeat Figures 3C and 3D and include simultaneous transfection with P1-3Cwt (None/P1-3Cwt, VP1/P1-3Cwt and 2B/P1-3Cwt) at the same time.

Line 200: Replace “for” with “from”.

Author Response

The authors should clarify the method used for luciferase assays. The authors state in the materials and methods section that “Luciferase assays were performed as previously described [15].” This reference states “Luciferase activity was detected in media separated from transfected cells”.  Was this the method used?

  • Yes, the luciferase is rapidly secreted into cell culture media and is remarkably stable. For the assay we briefly centrifuge harvested cell culture media in order to pellet cellular material.  The listed reference, Martel et al 2019, also describes the luminometer utilized, volume of diluted cell culture media utilized, substrate utilized, and integration times.  The authors have expanded upon the methodology in Section 2.3.

With regards to the data presented in Figure 1B, is it possible that expression of wild type 2B affects GLuc (transgene) secretion into the medium rather than expression per se? Why does the western blot of cell lysates in Figure 5A show comparable GLuc expression for both P1-L127P or P1-2B-L127P? – these results appear inconsistent with those presented in Figure 3B which show “a reduction in transgene expression with inclusion of 2B”. Likewise, in Figure 6 the amount of GLuc expressed in the absence or presence of 2B appear comparable when cell lysates are analysed by western blot.

  • The goal of Figure 5 is to demonstrate that we were unable to detect extracted antigen utilizing our established procedures. As the overall expression of 2B containing constructs with the L127P protease was lower we had to account for the possibility that detection was hindered simply by decreased expression from the constructs.  To account for this we equalized loading based off GLuc levels, expressed at the same time as the antigen, which we could confirm by both activity assay and western blotting. The same is true for Figure 6 which sought to evaluate modified lysis buffers to determine if extraction could be enhanced.

The authors show that the expression of 2B prevents antigen extraction using the listed lysis buffers. Have the authors analysed non-clarified whole cell lysates to determine if the antigen is pelleted with the cellular debris when 2B is expressed?

  • The authors utilized the Qproteome cell compartment kit (Qiagen) to identify if antigen was insoluble. This protocol includes extraction utilizing the supplied buffer CE4 which solubilizes all residual proteins which includes the cytoskeletal fraction.  When utilizing the P1-L127P plasmid most antigen is present in the Cytosolic fraction with residual antigen in the Membrane and Cytoskeletal fractions.  When utilizing the P1-2B-L127P plasmid no antigen was readily detected in any fraction.  An image of the western blot is included as an attachment.

Line 153: The authors should clarify the function of the additional sequences derived from FMDV non-structural proteins 2C, 3A, and 3B.

  • The molecular function, if any, of the additional sequences is unknown. It is hypothesized that it provides a site for 3C cleavage between 2B and 3C however the authors are unaware of definitive evidence confirming that processing at these sites occurs.  The inclusion of these sequences in Ad5 constructs pre-dates widely available gene synthesis and cloning technologies.  The sequences were included in the original Ad5 vaccine constructs as a carryover from cloning into the Ad5 vector requiring the utilization of available cut sites for insertion.  As their removal would constitute a change to the constructs from a regulatory standpoint it would invalidate all previous data and require repeating of a large number of animal studies involving efficacy and safety.  Subsequently, they have been maintained in many Ad5 and other vaccine constructs.

Line 160, and Figures 3A and 3B: It’s difficult to compare the results presented in Figures 3A and 3B if they were not carried out at the same time. The authors should repeat Figures 3C and 3D and include simultaneous transfection with P1-3Cwt (None/P1-3Cwt, VP1/P1-3Cwt and 2B/P1-3Cwt) at the same time.

  • The experiments reported in Figure 3A and 3B were carried out at the same time however they were split into separate graphs to allow the reader to better interpret the data presented with constructs containing 3Cwt. There is over a log difference in expression between P1-L127P and both P1-3Cwt and P1-2B-3Cwt. In previously published work, Puckette et al 2017, the authors have observed a similar difference in expression between 3Cwt and 3CL127P constructs making comparisons on the same graph scale difficult. Switching to a logarithmic scale however makes visualization of the differences between P1-3Cwt and P1-2B-3Cwt as well as the various 2B mutants with L127P more difficult.  The authors believe that separating the data allows for the results to be better visualized to the reader, that inclusion of 2B enhances a 3Cwt construct while detracting from that of L127P.

Line 200: Replace “for” with “from”.

  • Changed

This manuscript is a resubmission of an earlier submission. The following is a list of the peer review reports and author responses from that submission.

Round 1

Reviewer 1 Report

Comments for the Author:

Foot-and-mouth disease virus (FMDV) is a highly contagious virus affecting cloven-hoofed animals, including cattle, pigs, and sheep. The development of FMD VLP vaccine is significant in preventing and controlling FMD. Previously, it has been demonstrated that incorporation of 2B into an FMD molecular vaccine construct containing a wildtype FMDV 3C protease resulted in enhancement of transgene expression. In this work, the authors showed that the benefit to transgene expression was negated when using the FMDV 3CL127P mutant, which has reduced processing of host cellular proteins, a reversion resulting from 2B viroporin activity. This has implications for future molecular FMD vaccine constructs, which may utilize mutant FMDV 3C proteases. Overall, this is an advance in FMDV vaccine development.

The methods and the data presented look convincing and well-presented. However, some points are needed to be addressed before publication.

I recommend the following changes:

Minor issues:

1) Adding FMDV infected samples as positive control would be better for Figure 5 and Figure 6.

2) Could the VLP arrays within cells transfected with plasmids P1-L127P be purified and observed by TEM? 

Reviewer 2 Report

In this manuscript, Primavera et al. evaluated the effects of Foot-and-mouth disease(FMD) 2B viroporin when incorporated into a plasmid-based virus-like particle (VLP). The authors have a lot of experience in this field, and this manuscript provides information that could help design future vaccines against FMD. 

The manuscript is well written. The experiments are well designed. Results are clear and concise, and the discussion focuses on the main points of the manuscript.   

As the authors mentioned, one of the most important questions for this reviewer is how these changes modify the immunity induced in animals. In general, I think that is a manuscript with the merits to be published, but I am uncertain if Viruses is the best option, considering the high impact factor of this journal. I am sure that the Editor has a better opinion than me in this sense. 

Reviewer 3 Report

The manuscript by Primavera et al., describes studies on the effect of the foot-and-mouth disease virus (FMDV) 2B protein on the production of proteins within transfected cells. This study follows on from earlier studies that reported an effect of the FMDV 2B on the efficacy of adenovirus vectors that express FMDV virus like particles (VLPs). In the current study, the effect of 2B (and mutant forms of it) in conjunction with wt and mutant forms of the FMDV 3C protease have been analyzed. It appears that multiple mechanisms may underlie the changes in protein expression observed.  However, these mechanisms are not well defined. Importantly, experiments often lack certain appropriate controls. The presentation of the manuscript could also be significantly improved in many ways, e.g. annotation of Figures.

Specific points:

1)  In the Materials and Methods section, many details of the constructs are lacking, e.g. presumably initiation and termination codons were added. The Figures describing the constructs, Fig. 1A and Figure 2, are very simplistic and do not provide much additional information. It would be much better if the structures of the plasmids used in the experiments that produced data (Fig 1B, Fig. 3, Fig.5) were all shown and labelled in the same way.

2) The labelling on the y axis of the graph in Figure 1B makes no sense, indeed 2 x 108 appears twice!

3) It seems surprising that the 2BΔAB has more of an inhibitory effect that either the 2BΔA or 2BΔB mutants alone (see Figure 1B). Why should this be? The experiment should also include a GLuc2A construct without any 2B sequences, to give the maximal value for the Gluc expression.

4) In Figure 3C, the experiment is supposed to show the effect of co-transfection but in one of the 3 cases no second plasmid is used. It would be better if the empty vector was used as the second plasmid.

5) The images shown in Figure 4 are not very compelling in themselves. There appears to be no indication of which cells were used for this experiment and there is no negative control (untransfected). There should also be some means of showing that the arrays are indeed FMDV antigen, e.g. using immunolabelling. Otherwise, I think it should be deleted.

6) In Figure 5A, very similar levels of Gluc are observed by WB from both P1-L127P and P1-2B-L127P, is this because of the normalization of the loading for Gluc expression (as indicated in lines 124-125)? It could be expected that the 2B would inhibit expression of the Gluc too (see Fig 3B). How does this fit with the comment on lines 213-215? It would be better to load equal amounts of the cell lysates and to detect a cell marker, e.g. actin or tubulin to show similar protein loading.

Minor points:

a) Line 34, the FMDV genome does not “encode a single open reading frame including structural proteins…”. An open reading frame is a property of the RNA and is included within it. The genome sequence encodes a polyprotein that includes structural.. and non-structural proteins….

b) Line 37. Currently, this sentence makes no sense, it should indicate that intact capsid structures are required to generate protective immunity.

c) Lines 37-38 and elsewhere. Personally, I do not like the terminology “molecular vaccines” or “molecular FMD vaccines”. Are there any vaccines that are not made of molecules?

d) Subscripts should be used appropriately, e.g. CO2 should be CO2 and MgCl2 should be MgCl2.

e) The legend for Figure 2 is poor and does not match the Figure itself.